# Biopolymer-Based Composites: An Eco-Friendly Alternative from Agricultural Waste Biomass

**Yashas Gowda T. G.** [1], **Sharath Ballupete Nagaraju** [1], **Madhu Puttegowda** [1,*], **Akarsh Verma** [2,3,*], **Sanjay Mavinkere Rangappa** [4] and **Suchart Siengchin** [4]

1. Department of Mechanical Engineering, Malnad College of Engineering, Hassan, Visvesvaraya Technological University, Belagavi 573202, India; yashasmce@gmail.com (Y.G.T.G.); sbn@mcehassan.ac.in (S.B.N.)
2. Department of Mechanical Science and Bioengineering, Osaka University, Osaka 560-8531, Japan
3. Department of Mechanical Engineering, University of Petroleum and Energy Studies, Dehradun 248007, India
4. Natural Composites Research Group Lab, Department of Materials and Production Engineering, The Sirindhorn International Thai-German Graduate School of Engineering (TGGS), King Mongkut's University of Technology North Bangkok (KMUTNB), Bangkok 10800, Thailand; mcemrs@gmail.com (S.M.R.); suchart.s.pe@tggs-bangkok.org (S.S.)
\* Correspondence: pm@mcehassan.ac.in (M.P.); akarshverma007@gmail.com (A.V.)

**Abstract:** This review article addresses the potential for biopolymer-based composites made from agricultural waste biomass to replace conventional materials in a sustainable and responsible manner. The composition and manufacturing method of biopolymer-based composites are described in the article, along with some of their distinctive qualities and benefits, such as their low cost, renewable nature, and biodegradability. The article also shows a number of real-world uses for these composites, including packaging, construction, vehicle parts, biofuels, soil amendments, and medical uses. Overall, the article highlights the potential of biopolymer-based composites made from agricultural waste biomass for lowering waste generation, decreasing dependency on non-renewable resources, and boosting sustainability in a variety of industries.

**Keywords:** biopolymer-based composites; agricultural waste biomass; sustainable materials; industrial applications; environmental impact

## 1. Introduction

Natural fibers are derived from a variety of flora and fauna sources and are utilized in the production of textiles and other commodities. These materials are recognized for their capacity to decompose naturally, their ability to endure over time, and their positive impact on the ecosystem. Cotton, wool, silk, flax, and jute are among the frequently utilized natural fibers. Because of their exceptional qualities, including suppleness, breathability, and longevity, these fibers are ideally suited for usage in a wide range of different applications. Moreover, natural fibers possess the quality of renewability rendering them a more ecologically conscientious alternative. Notwithstanding their benefits, natural fibers may exhibit certain limitations, including susceptibility to light and moisture, as well as reduced tensile strength when compared to synthetic fibers [1–3]. In spite of these constraints, the utilization of organic fibers is progressively expanding owing to their favorable ecological influence and widespread acceptance among customers. The term "natural fibers" refers to fibers that have been procured from agricultural biomass, which includes both crop residue and animal fibers, such as wool, silk, chicken feather, and hair. These particular fibers are deemed ecologically sustainable due to their composition from renewable resources and the absence of hazardous chemicals in the manufacturing process. Several natural fibers can be derived from agricultural biomass, such as jute, sisal, coir, bamboo, and hemp [4–6]. The described fibers are utilized in the production of various commodities, including but not limited to textiles, rope, mats, and paper. A variety of properties are

provided by these materials, including but not limited to high tensile strength, durability, and biodegradability. Moreover, the utilization of these fibers aids in mitigating the surplus generated from agricultural production, rendering it a sustainable and environmentally conscious substitute to synthetic fibers. In general, the utilization of natural fibers derived from agricultural biomass presents a potentially viable approach towards mitigating the ecological ramifications associated with the textile and manufacturing sectors. The utilization of agricultural waste biomass to create biopolymer-based composites is a promising approach to developing composite materials. These composites incorporate plant-derived biopolymers, including starch, cellulose, and proteins, as a means of reinforcing a polymer matrix [7,8]. The fact that these composites are both renewable and biodegradable makes them a more environmentally friendly choice than traditional composites that are based on petroleum. Starch, cellulose acetate, and chitosan are examples of some of the biopolymers that are utilized often. The composites under consideration incorporate agricultural waste biomass, comprising residual materials from crops, including but not limited to straw, stalks, leaves, and husks. The previously mentioned substances possess the potential to undergo processing and subsequent conversion into biopolymers, which are subsequently employed as a means of reinforcing the polymer matrix. When compared to conventional composites, lightweight biopolymer-based composites manufactured from agricultural waste biomass provide various benefits over their more conventional counterparts, including a lower environmental impact, cheaper costs, and superior mechanical qualities. These composites have a wide range of potential uses, including lightweight applications in the automobile industry, the packaging industry, and the construction industry [9]. Because of their lower density and inherent sustainability, they present an intriguing possibility as a means of lowering the carbon footprint left by the industrial sector [10,11]. Further study is required to enhance their capabilities and qualities before they can be considered as a viable option for use in commercial applications. The utilization of agricultural waste biomass in the fabrication of composites represents a sustainable and ecologically sound methodology to produce composite materials. These materials can undergo processing to serve as reinforcement in a polymer matrix. The consumption of agricultural waste biomass as a reinforcement agent in composite materials serves to mitigate the amount of waste produced from agricultural activities, while concurrently diminishing reliance on non-renewable resources. Composites derived from agricultural waste biomass and composed of biopolymers exhibit considerable potential as a means of mitigating the ecological footprint of the composite sector. Composites possess versatile applicability, including but not limited to employment in the automotive industry, packaging sector, and construction materials [12–15] (refer to Figure 1 [15]). The comparatively lower cost and biodegradable properties of these materials render them an eco-friendlier substitute for conventional composites. Furthermore, making use of these products may result in the emergence of novel commodities and industries, thereby generating economic advantages for agricultural producers and local societies. Nonetheless, the utilization of agricultural waste biomass in composite manufacturing is subject to certain constraints, including inconsistencies in the waste's quality and accessibility, as well as processing complexities. However, through additional investigation and advancement, the utilization of agricultural waste biomass in the production of composites holds promise in promoting a more ecologically conscious and sustainable future.

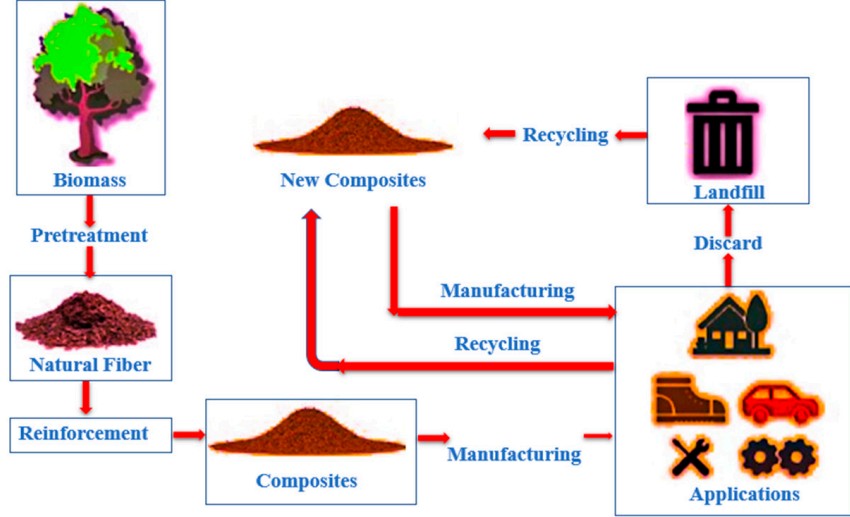

**Figure 1.** Converting biomass to composites [15].

## 2. Fibers from Agricultural Wastes

Fibers derived from agricultural waste biomass pertain to fibers procured from the residual components of crops, including but not limited to straw, stalks, leaves, and husks. Composites, nonwoven, and biodegradables can all benefit from the use of these fibers as reinforcement thanks to their easy extraction and processing. The use of fibers derived from agricultural waste biomass has environmental advantages, such as the fact that they are renewable and biodegradable. In addition to fostering a more sustainable future, the use of these fibers can help to reduce dependency on finite resources [16,17]. Notwithstanding, the utilization of said fibers is accompanied by certain obstacles, including inconsistencies in the caliber and accessibility of the refuse, alongside processing intricacies. However, through additional investigation and advancement, fibers derived from agricultural waste biomass possess the capability to emerge as a prevalent and significant constituent of a sustainable and eco-friendly prospect. Nonwoven materials and biodegradable goods made from a variety of fibers that may be utilized as reinforcement in composite materials are one way that people are making strides towards a greener tomorrow [18–21] (refer Figure 2 [19] for different types of agricultural wastes). Figures 3 and 4 [20] showcases density (with percentage of elongation) and mechanical properties [21] of fibers from agricultural wastes, respectively.

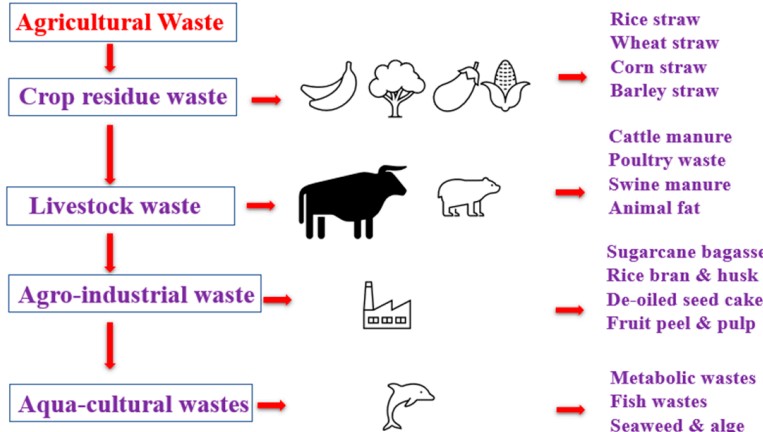

**Figure 2.** Types of agricultural wastes [19].

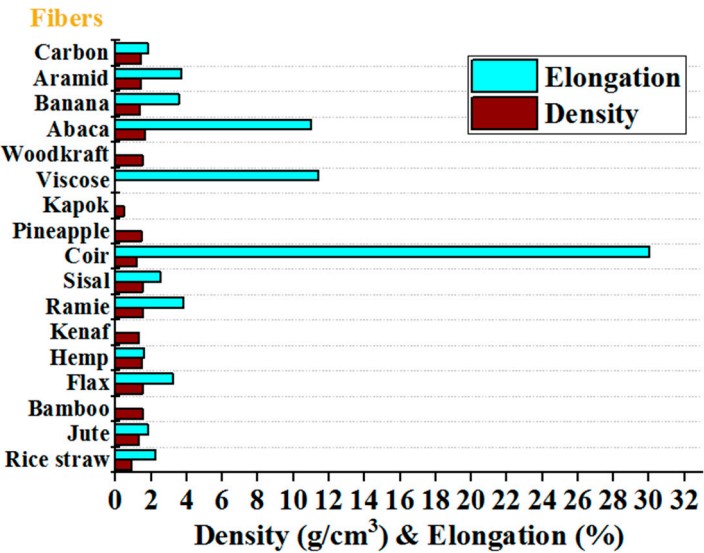

**Figure 3.** Density and percentage of elongation of fibers from agricultural wastes [20].

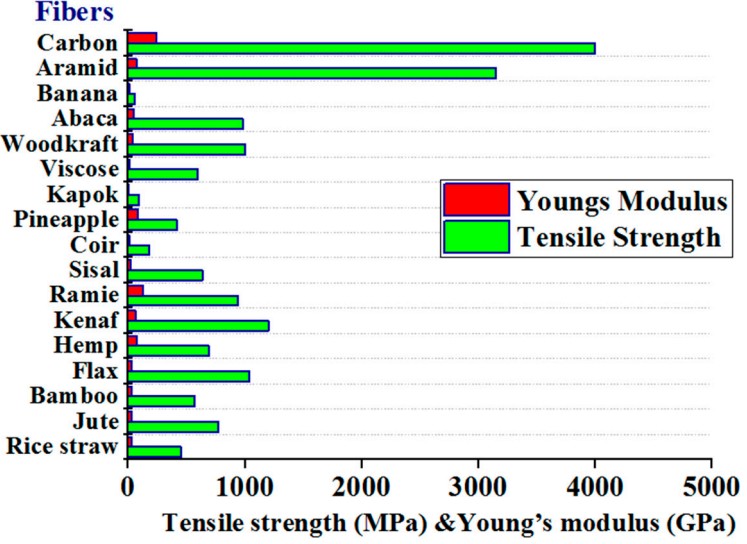

**Figure 4.** Tensile strength and Young's modulus of fibers from agricultural wastes [21].

### 2.1. Rice Straw Fiber

When rice is harvested, a considerable quantity of rice straw is produced as a byproduct. To decrease waste and boost sustainability in the composites, rice straw fibers have become increasingly popular as an additive in recent years. Since rice straw fibers are inexpensive, abundant, and biodegradable, they constitute an excellent material for composites. The incorporation of rice straw fibers as a reinforcing agent in biopolymer composites has been found to enhance their mechanical and thermal characteristics, thereby rendering them more appropriate for diverse applications [22–25]. Rice-straw-fiber-reinforced composites are being formulated and utilized in diverse fields, such as the automotive industry, construction sector, and packaging industry. Because it creates a product with additional value out of something that was previously thought of as waste, the use of rice straw fiber in composites can also generate economic advantages for local communities and farmers [26]. Notwithstanding, the advancement of composites based on rice straw fiber remains a growing field of inquiry, and there exist various obstacles that require resolution. The challenges encountered in this context pertain to the variability in the quality and accessibility of rice straw fibers, alongside the intricacies involved in processing them. Despite the current limitations, through additional investigation and advancement, the

utilization of rice straw fibers in composite applications exhibits promising prospects for a more ecologically conscious and sustainable future [27–29].

### 2.2. Bamboo Fiber

Bamboo fiber is a naturally occurring fiber that is derived from the bamboo plant. As a eco-friendlier and more long-lasting substitute to synthetic fibers, its appeal has risen in recent years. Bamboo fibers are versatile due to their great mechanical strength, high resilience, and ability to absorb and release moisture. The incorporation of bamboo fibers as reinforcement in biopolymer composites has been found to enhance the mechanical characteristics of the composite, thereby rendering it more appropriate for diverse applications. Due to its light weight, low cost, high strength, and rigidity, bamboo fiber is used as reinforcement in polymeric materials. Historically, bamboo has been used to construct dwellings, bridges, and traditional boats [30–32]. The utilization of bamboo fiber in composite materials exhibits promising prospects for offering a sustainable and ecologically sound substitute to synthetic fibers. Bamboo-fiber-reinforced composites have the potential to serve as a viable and eco-friendly substitute for conventional construction materials, including wood and concrete. Additionally, they can be utilized to fabricate lightweight and resilient components for the automotive sector, as well as for packaging purposes. Offering an eco-friendly and decomposable substitute for conventional packaging elements, athletic equipment, including skateboards and surfboards [33,34], bamboo fibers possess inherent softness and effective moisture absorption characteristics, rendering them a viable option to produce environmentally sustainable textiles and apparel. Alternatives to conventional composite materials that are more ecologically responsible and sustainable might be made possible by using bamboo fiber in composites. Despite that, additional investigation and advancement are required to completely actualize the potential of bamboo fiber in composite applications [35,36].

### 2.3. Bagasse

Bagasse refers to the residual fibrous material that remains after the extraction of juice from sugarcane. Nevertheless, additional investigation and advancement are required to completely actualize the potential of bagasse fiber in composite applications. It is a form of biomass that comes from farms and has been investigated as a natural fiber option for composites [37]. Bagasse fiber is a good choice for composites because it has many good qualities. For example, bagasse is a waste of sugarcane processing and is made in large quantities all over the world, making it a cheap and easy-to-obtain source of fiber. Researchers have been able to exploit this biomass for a variety of applications, including energy and environmental sustainability. The low density of bagasse fiber renders it a desirable candidate for reinforcing lightweight composites [38–41]. Bagasse fiber exhibits favorable mechanical properties, including notable tensile strength, stiffness, and durability, rendering it a viable candidate for use as a reinforcing agent in composite materials. Bagasse fiber exhibits biodegradability and eco-friendliness, rendering it a viable substitute for synthetic fibers in composite applications [42–44]. Numerous industries have experimented with bagasse-fiber-reinforced composites, and some of them include construction, packaging, and automobiles. On the other hand, additional research is required before bagasse fiber composites may be used in commercial applications.

### 2.4. Banana Fiber

Banana fiber is a natural fiber derived from the stem and pseudo-stem of the banana plant. It is a substance that is regarded to be agricultural waste and is generated in vast amounts in many nations, particularly those countries where bananas are farmed for commercial purposes. Banana fiber possesses several noteworthy characteristics and practical uses [45,46]. The tensile strength of banana fiber is noteworthy, and its high moisture resistance renders it appropriate for diverse applications. Additionally, it possesses biodegradable property and is conducive to environmental sustainability. The utilization

of banana fiber is observed in the manufacturing of various textile products, including but not limited to fabrics, ropes, and mats. The material in question is recognized for its characteristics of being pliable, long-lasting, and resilient against deterioration [47,48]. The utilization of banana fiber as a reinforcing agent in composite materials has been investigated, including its application in biodegradable plastics, natural rubber composites, and bamboo composites [49]. Composites possess potential utility in scenarios where there is a preference for materials that are both lightweight, such as in the fabrication of automotive components and construction materials. The uses of banana fiber extend beyond its conventional applications, as it can also serve as a viable material in the manufacturing of paper and various other commodities, including but not limited to towels, napkins, and tissues [50–52]. The use of banana fiber in different industries offers both academics and businesses a new way to think about how it could be used in the future. The fiber content and qualities of strength are the major issues that affect whether banana fiber can be used for certain purposes [53].

### 2.5. Kenaf Fibers

The distinctive attributes and qualities of kenaf fibers render them appropriate for diverse applications in the composites and other industrial sectors. Since kenaf can be grown in many different climates and countries, it can help to lessen the world's reliance on non-renewable resources, such as fossil fuels. Kenaf fibers are eco-friendly since they are fully biodegradable. These fibers were once utilized to make fabrics, cords, ropes, storage bags, and boats by the Egyptians. These fibers are at present manufactured as composites with other materials and utilized in automotive, construction, packaging, furniture, textiles, matting, paper pulp, and other applications [54–56]. The fibers derived from kenaf exhibit a low density of approximately $1.3 \text{ g/cm}^3$, rendering them appropriate for deployment in lightweight applications. The thermal stability of kenaf fibers is noteworthy as they demonstrate resilience against thermal degradation when exposed to elevated temperatures. Because kenaf fibers absorb very little water, they are not easily degraded by exposure to damp conditions [57–60]. The chemical resilience of kenaf fibers makes them useful in industrial settings where toxic substances are present. Easy to spin into yarns or weave into fabric, kenaf fibers have several potential uses [60].

### 2.6. Jute Fiber

Jute cultivation is primarily carried out in favorable agro-climatic conditions, predominantly in the Bengal delta region spanning across India and Bangladesh. The crop is harvested upon attaining maturity, which generally takes around 4–5 months. The jute stems that have been collected are subjected to a process called retting, in which they are immersed in water for a prolonged period to facilitate the decomposition of non-fibrous components and to render the fibers more pliable [61,62]. Raw jute fibers are obtained by manually or mechanically stripping the fibers from the stems. The process of scotching is employed on the raw jute fibers to mechanically extract the fibers from any residual non-fibrous components. Subsequently, the jute fibers are subjected to a drying process to eliminate any residual moisture. The retting, stripping, scotching, and drying procedures are used in conjunction with one another in order to remove the jute fibers from the stems of the jute plant. They are processed to remove contaminants and boost mechanical qualities, such as the tensile strength and the Young's modulus. The potential treatment modalities encompass chemical interventions, such as bleaching, or physical interventions, such as heat treatment [63,64]. Fibers find application in diverse fields, including but not limited to textiles. Jute fiber is utilized in the production of diverse textiles, such as hessian and jute sacking. Jute fiber has been utilized in the production of eco-friendly packaging materials, including bags and sacks that are capable of decomposing naturally [65–67]. Reinforcing the structure of building materials, such as cement boards, floor tiles, and non-woven geotextiles, is one of the many applications for jute fiber in the building and construction industry. Due to its ability to provide both acoustic and thermal insulation, jute fiber is

frequently utilized in the automobile industry as a reinforcing material in components, such as door panels and trunk liners [67].

### 2.7. Hemp Fiber

Hemp fiber composites are materials in which the matrix is made of synthetic or biodegradable polymers, with hemp fibers acting as the main reinforcement component. Due to their excellent mechanical properties, including high tensile strength and Young's modulus as well as notable stability, synthetic polymers such as polypropylene (PP), polyethylene (PE), and polyvinyl chloride (PVC), have received extensive research as matrices for hemp fiber composites [67,68]. These materials are not eco-sustainable due to the fact that they are not biodegradable. On the other hand, due to their promising outcomes in terms of mechanical characteristics, biodegradability, and sustainability in recent years, starch-based polymers have attracted substantial attention as a biodegradable matrix for hemp fiber composites. The addition of hemp fibers as reinforcement has further improved these composites' properties, making them suitable for a variety of uses in the automotive, construction, and packaging industries [69]. It should be emphasized that the biodegradability of hemp fibers is favorable in biodegradable matrixes, but not in matrixes from petroleum-based polymers, when comparing the characteristics and economy of utilizing hemp and conventional fibers as fillers for petroleum-based polymers. Therefore, when reinforced with hemp fibers, it is important to compare the mechanical characteristics, biodegradability, and sustainability of biodegradable matrixes, such as starch-based polymers and petroleum-based polymers [70,71]. The choice of a practical biopolymer matrix for composites made of hemp fibers depends on the application and the desired balance between effectiveness and environmental effects. Polyhydroxyalkanoates (PHA), cellulose-based polymers, and polylactic acid (PLA) are a few examples of biopolymer matrices for hemp fiber composites. These biopolymer matrices have respectable mechanical characteristics and outstanding biodegradability and sustainability [72]. Regardless of the type of matrix used, whether it be synthetic or biodegradable polymers, hemp fiber composites display desirable mechanical characteristics and stability. Biodegradable matrixes, such as starch-based polymers or biopolymers, are more environmentally benign and better suited for usage where the final product will be discarded when taking the ecological effects of the composite into account.

### 2.8. Sisal Fiber

The sisal plant, known for its ability to withstand drought, is cultivated in tropical and subtropical areas, with a primary focus on nations such as Brazil, Tanzania, and Kenya. Sisal plants are typically propagated using cuttings or tissue culture techniques. Once established, these plants undergo a growth period of roughly 2–3 years, during which they attain maturity and become suitable for harvesting. The manual harvesting of sisal leaves typically occurs at intervals of 9–12 months [73–75]. The process of obtaining sisal fibers involves a sequence of procedures, such as stripping off the leaves, purification, and desiccation. The process of retting involves immersing the fibers in water to facilitate their decomposition and subsequent separation from the leaf pith. The process of retting plays a crucial role in the extraction of sisal fibers, as it effectively segregates the fibers from the pith and any other non-fibrous constituents. Retting is a process that can be carried out through the utilization of various agents, such as fresh water, seawater, or chemical substances. As a biopolymer, sisal fiber has several potential applications. The mechanical characteristics of bio-composites made from sisal fiber and biodegradable polymers, such as polylactic acid (PLA) or starch, can be significantly enhanced [76,77]. Geotextiles can be reinforced with sisal fibers to enhance their strength and durability. Sisal fibers have the potential to serve as a viable source to produce biodegradable packaging materials, including but not limited to bags and sacks. Sisal fibers possess favorable sound and thermal insulation characteristics, rendering them suitable for deployment as reinforcement in automotive components, including door panels and trunk liners. Sisal fibers have been identified as

a viable option for reinforcing automotive components, including door panels and trunk liners, owing to their favorable sound and thermal insulation characteristics [78,79].

*2.9. Abaca Fiber*

Abaca fiber composites refer to composite materials that utilize abaca fibers as a reinforcing agent, in conjunction with a polymer matrix. Several advantages can be gained by using abaca fibers into composites. An eco-friendly substitute for conventional fiber reinforcements, such as glass and carbon fibers, abaca is a renewable material farmed mostly in the Philippines. Because of their great tensile strength, abaca fibers are ideally suited for usage in high-strength composites in fields such as structural engineering [80]. The chemical resistance of abaca fibers has been demonstrated, rendering them appropriate for utilization in scenarios where chemical exposure is a potential issue. The biodegradability of abaca fibers renders them a viable alternative for use in scenarios where the eventual disposal of the product is necessary. Since abaca fibers seem natural, they are a good choice for uses where visual appeal is paramount, such as in the automobile and interior design sectors [81–83]. Abaca fiber composites have several advantages over composites created from other materials, but their manufacturing is hindered by the high cost of production and the scarcity of good-quality abaca fibers. New methods of abaca fiber extraction and processing are at present being developed to address this issue, with the hopes of increasing the supply of high-quality fibers while decreasing manufacturing costs [84].

## 3. Composition of the Natural Fibers from Agricultural Waste Biomass

Agricultural waste biomass consists of plant matter that has been harvested but is otherwise destined for disposal as a byproduct of farming. Natural fibers extracted from biomass waste generated in agriculture might differ substantially in composition. However, typically, these components comprise the following. Cellulose constitutes the primary constituent of most natural fibers, comprising approximately 30–50% of fiber. The strength and stiffness of the fiber is attributed to the presence of cellulose [85,86]. Hemicellulose constitutes approximately 10–30% of the fiber and imparts elasticity to it. Lignin, a constituent of plant cell walls, constitutes a significant proportion of the fiber ranging from 10–30% and confers hydrophobic properties to the fiber. Pectin is a constituent of dietary fiber that plays a minor role in the structural integrity of fibers by promoting their cohesion. Waxes and oils are minor constituents that offer hydrophobic characteristics to the fiber [84,86]. The composition of this constituent comprises minerals and other inorganic substances, and its proportion in the fiber varies based on the origin of the agricultural waste biomass. Agricultural waste biomass may comprise impurities, such as dirt and sand, which have the potential to impact the mechanical and physical characteristics of the fibers. Natural fibers extracted from biowaste can have a wide range of compositions depending on the species, growing circumstances, and processing methods employed. Because of how this variance might alter the fibers' mechanical and physical characteristics, it is crucial to precisely regulate the composition of these fibers for targeted uses [87,88]. Agricultural waste biomass has the potential to be employed as a reinforcing element in polymer composites, which would result in the production of ecologically benign and sustainable composite materials. However, there are a number of obstacles to overcome when using waste biomass from agriculture as reinforcement in polymer composites, including compositional, structural, and property variation in agricultural waste biomass, which makes it challenging to manufacture uniform composite materials [88]. The processing of agricultural waste biomass may pose difficulties owing to its variability, thereby presenting challenges in the production of fibers and composites of superior quality. It is challenging to create composites with desirable mechanical and physical characteristics due to the incompatibility of agricultural waste biomass with the polymer matrix. Competitiveness is hindered by the fact that biomass from agricultural waste is sometimes more expensive than more conventional fiber reinforcements, such as glass and carbon fibers [85,87–89]. Research is continuing to find ways to improve the

compatibility of agricultural waste biomass with polymer matrices, lower manufacturing costs, and address these other issues. Polymer composites that include agricultural waste biomass as reinforcement may also have positive effects on the environment, making them a desirable choice for eco-conscious customers and industries [89]. Figure 5 [90] reflects on the composition of the natural fibers from agricultural waste biomass.

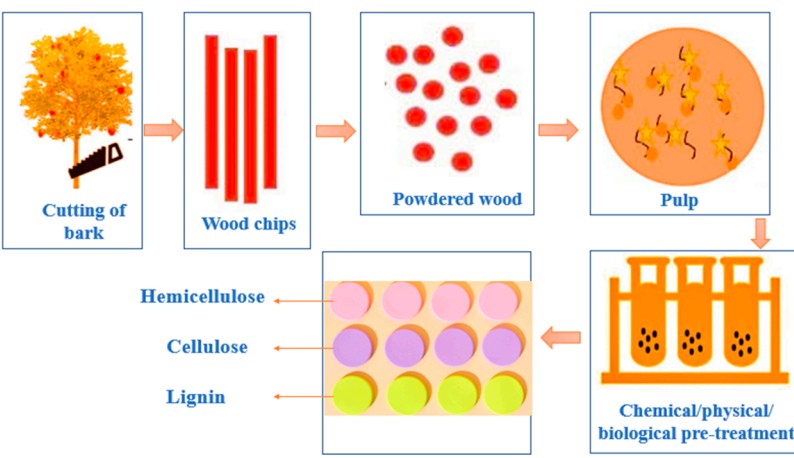

**Figure 5.** Composition of the natural fibers from agricultural waste biomass [90].

## 4. Chemical Treatments

Biomass from agricultural byproducts can be chemically modified to enhance its characteristics and increase its potential for usage as a reinforcing agent in polymer composites. The most often used chemical treatments are discussed in the Figure 6 [90].

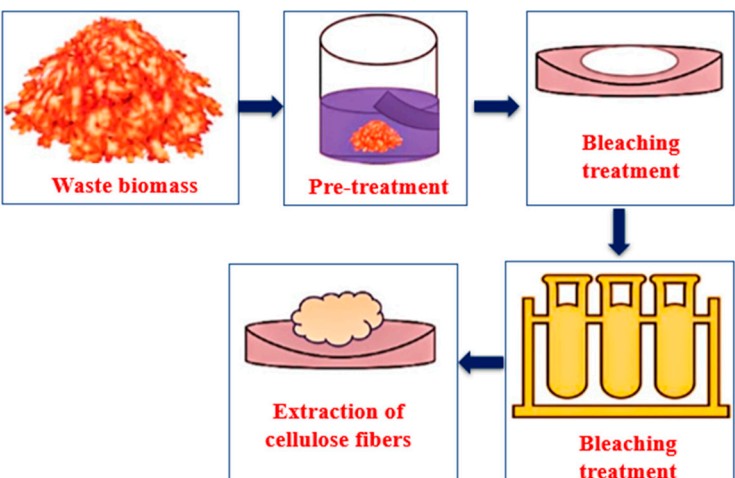

**Figure 6.** Chemical treatments [90].

### 4.1. Acid Treatment

The utilization of acid treatment has been observed to effectively eliminate lignin and hemicellulose from fibers, thereby enhancing their compatibility with polymer matrices. The acid treatment method is a chemical procedure that is employed to alter the characteristics of biomass fibers derived from agricultural waste [91]. This is performed to enhance their suitability as reinforcements in polymer composites. The procedure entails subjecting the fibers to an acidic solution, commonly composed of sulfuric or hydrochloric acid, with the aim of eliminating lignin and hemicellulose from the fibers. Acid treatment can improve fiber-to-matrix adhesion by eliminating lignin and hemicellulose, making the fibers more compatible with polymer matrices. The mechanical properties of fibers

can be enhanced through acid treatment, resulting in increased strength and durability, by eliminating impurities. The application of acid treatment has the potential to enhance the fibers' whiteness, thereby augmenting their aesthetic appeal for deployment in composite materials [92–94]. Nevertheless, acid treatment may entail certain drawbacks that could pose a threat to the environment if not adequately handled and eliminated. The degradation of fibers resulting in diminished strength and durability can also be observed as a consequence of acid treatment. The cost of acid treatment can be substantial, especially when utilizing acids of elevated purity [88]. In general, the utilization of acid treatment can prove to be a viable approach for altering the characteristics of agricultural waste biomass fibers, thereby rendering them suitable for employment in polymer composites. However, it is imperative to exercise prudence while assessing the ecological ramifications and financial implications of this treatment prior to its implementation [94,95].

### 4.2. Alkali Treatment

The application of an alkali treatment has been demonstrated to effectively eliminate hemicellulose from fibers, leading to a notable enhancement in their mechanical characteristics. The impact of an alkali treatment on biomass fibers derived from agricultural waste is contingent upon the fiber type and the specific treatment conditions employed. The following are some of the general outcomes that can occur when biomass fibers from agricultural waste are treated with alkali [96]. The removal of hemicellulose from fibers through an alkali treatment has been observed to enhance their mechanical characteristics, resulting in increased strength and durability. The application of an alkali treatment to fibers can enhance their compatibility with polymer matrices, leading to improved interfacial adhesion between the fibers and matrix. The application of an alkali treatment has the potential to enhance the surface area of fibers, thereby augmenting their efficacy as reinforcements in composite materials [97,98]. The application of an alkali treatment has been observed to result in a reduction in the moisture absorption capacity of fibers. This phenomenon has been found to positively impact the dimensional stability of the fibers, thereby mitigating the occurrence of swelling and shrinkage during usage. The implementation of an alkali treatment has the potential to decrease the surface roughness of fibers, thereby enhancing their processability by rendering them smoother [99,100]. It is noteworthy that the application of an alkali treatment may result in detrimental outcomes on the fibers, including fiber degradation and weakened strength. This is especially true if the treatment conditions are not meticulously regulated.

### 4.3. Bleaching

In composite materials, natural fibers, such as cotton, linen, and silk, are frequently employed as reinforcing elements. The inherent qualities of natural fibers, such as their high strength-to-weight ratio, biodegradability, and reliance on renewable resources, are combined with the strength and durability of synthetic resins in the creation of these materials. The bleaching process can have a considerable impact on the characteristics of natural fibers as well as the overall performance of the composite material [101]. The natural pigments included in the fibers can be removed during the bleaching process, which can cause a loss of color and sometimes strength. Furthermore, bleaching agents have the potential to weaken fibers, which would reduce their mechanical qualities, including stiffness and tensile strength. The type of fibers utilized, the type of bleaching agent employed, and the bleach concentration are only a few of the variables that affect how bleaching affects composite materials [102,103]. For instance, hydrogen peroxide, a milder bleaching solution, is less likely to harm natural fibers than chlorine bleach. Using the proper bleaching procedure and the circumstances for the fibers being used to reduce the effects of bleaching on composite products are important. To reduce the effect on the mechanical qualities of the fibers, this may entail employing lower bleach concentrations or bleaching for shorter periods of time [104]. To make up for any strength or other attributes lost because of bleaching, the composition of the composite material might also need to

be changed. In conclusion, the bleaching process can have a significant impact on the characteristics of natural fibers used in composite materials. To minimize the impact on the qualities of the fibers and the overall performance of the composite material, care should be taken when choosing the kind of fibers and bleaching chemicals [105].

*4.4. Silane Treatment*

Natural fibers used in composite materials often have their surfaces modified using silane. Silanes are a class of chemical compounds that include both organic and inorganic elements and are employed in composite materials to increase the adherence of the fibers to the resin matrix [106]. Applying a solution or vapor of silane compounds to the surface of natural fibers is known as a silane treatment. This solution is normally dissolved in a solvent, such as water or alcohol. The silanes then interact chemically with the fiber surfaces to create a connection between the fiber and resin matrix. The silane treatment has a substantial effect on natural fibers and composite materials. Strength, stiffness, and impact resistance are all increased mechanical qualities because of the silane coating's improvement in the compatibility of the fibers and resin [106–108]. The fibers are better suited for usage in outdoor or high-humidity applications thanks to the silane treatment's improved resistance to moisture and environmental deterioration. Silane treatment may influence how composite materials are processed. The process can enhance the resin matrix's flow around the fibers, resulting in greater impregnation and matrix-fiber adhesion. This may lead to a composite material with improved mechanical characteristics and greater homogeneity [109–112]. In conclusion, silane treatment is a useful technique for improving the surface of natural fibers used in composite materials. Improved mechanical capabilities and resistance to environmental deterioration result from the treatment's improvement of the compatibility between the fibers and the resin matrix. The processing of composite materials is significantly impacted by a silane treatment, which enhances impregnation and matrix–fiber adhesion.

**5. Biopolymer-Based Composites from Agricultural Waste Biomass**

The combination of biopolymers with agricultural waste biomass results in the creation of a new class of material known as biopolymer-based composites from agricultural waste biomass. Biopolymers refer to a class of naturally occurring polymers that are synthesized by living organisms, including but not limited to proteins, polysaccharides, and nucleic acids. Biomass derived from agricultural byproducts is known as "agricultural waste". This includes byproducts such as crop leftovers, sawdust, and animal dung [113,114]. In recent years, there has been a rise in interest in the creation of biopolymer-based composites made from agricultural waste biomass due to their promise as long-lasting and environmentally acceptable building materials. The packaging, building, and automobile sectors are only a few of the many possible uses for these composites. There are several stages involved in the production of biopolymer-based composites using biomass derived from agricultural byproducts. Initially, the biomass derived from agricultural waste is gathered and subjected to a purification process to eliminate any extraneous substances [115–117]. Subsequently, the biopolymer is combined with the agricultural biomass in a precise proportion, resulting in the creation of a composite material. Subsequently, the composite material is shaped into the intended form and subjected to a curing process to achieve the ultimate product. Biopolymer composites made from agricultural waste biomass have several benefits, including being inexpensive, renewable, and biodegradable. The productive use of these materials can potentially mitigate the ecological consequences of agricultural waste through its conversion into a valuable resource [118–120]. Nevertheless, the production of said materials presents certain challenges, including the requirement for processing conditions that are optimized and the variability of the biomass derived from agricultural waste. It can be stated that composites derived from biopolymers sourced from agricultural waste biomass present a viable and ecologically responsible substitute for conventional composite materials. Bio-polymers possess a diverse array of potential applications and have the ca-

pacity to mitigate waste production and reduce reliance on non-renewable resources [121]. Based on their source, biopolymers can be divided into three categories: those derived from plants, animals, and microorganisms. The distinctive characteristics of each type of biopolymer can make them appropriate for particular applications. For instance, because of their availability, affordability, and biodegradability, plant-based biopolymers, including cellulose, starch, and chitin, are frequently employed in the creation of biopolymer-based composites [122]. Biopolymer-based composites used in packaging, paper, and construction materials frequently contain cellulose, which is present in plant cell walls. Producing biodegradable plastics and packaging materials requires starch, which comes from crops such as corn and potatoes. Producing medical devices and bandages for wounds uses chitin, which comes from the shells of crustaceans. Gelatin and collagen are examples of animal-based biopolymers that are frequently employed in the food and pharmaceutical industries. Capsules, coatings, and gels are made with gelatin, which is obtained from animal bones and skin. Animal connective tissue contains collagen, which is used to make medical implants and bandages for wounds. Microorganisms can manufacture microbial-based biopolymers, such as polyhydroxyalkanoates (PHA), which are an alternative to petroleum-based plastics. PHA can be utilized in a range of applications, including packaging, agriculture, and medical devices. It is biodegradable and has strong thermal and mechanical qualities [123,124]. For instance, the business Ecovative Design creates biopolymer-based composites using mycelium, a fungus-based biopolymer, for insulation and packaging materials. Another illustration is NatureWorks, which creates polylactic acid (PLA)-based biopolymer-based composites for use in packaging, fibers, and other applications. Although there are drawbacks to using biopolymers in composites, such as their cost and processing requirements, their advantages, such as their biodegradability and renewability, make them a desirable choice for environmentally friendly industrial applications [125]. Figure 7 [90] depicts the biopolymer-based composites from agricultural waste biomass.

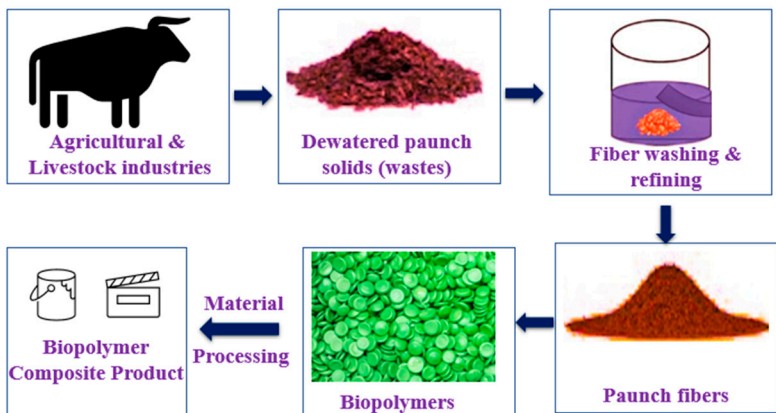

**Figure 7.** Biopolymer-based composites from agricultural waste biomass [90].

Due to their sustainability and environmental friendliness, biopolymer-based composites have been gaining appeal as an alternative to conventional synthetic composites. In this section, we compare biopolymer-based composites to conventional synthetic composites and explore their benefits and drawbacks. When disposed of, synthetic composites, which are frequently created from non-renewable materials and are not biodegradable, affect the environment [126,127]. Additionally, biopolymer-based composites outperform some conventional synthetic composites in terms of mechanical qualities, such as strength and stiffness. Additionally, a low-energy procedure can be used to create biopolymer-based composites, making them a more affordable choice. The limitations of the biopolymer-based composites are listed here. The difficulty of managing the characteristics of the materials is one of the key problems with biopolymer-based composites. Agricultural waste biomass's varied characteristics make it frequently challenging to build consistent

composites. Furthermore, the fabrication of biopolymer-based composites calls for particular processing conditions that might not always be practical or economical. Traditional synthetic composites, on the other hand, are more commonly used and understood in industry and may be made with consistent qualities [128–130]. Depending on the type of biopolymer and agricultural waste biomass used, as well as the processing parameters, the cost of making biopolymer-based composites can change. The long-term advantages of sustainability and biodegradability may make biopolymer-based composites, even though some of them may be more expensive to create initially, a more cost-effective option in the future. In addition, the production of biopolymer-based composites uses less energy than that of conventional synthetic composites, making them ultimately more economical and energy-efficient [131,132]. Overall, biopolymer-based composites outperform conventional synthetic composites in a number of ways, including sustainability, biodegradability, and superior mechanical qualities. Making biopolymer-based composites presents certain difficulties, however, due to the unpredictability of their characteristics and their particular processing needs. Nevertheless, biopolymer-based composites are a promising choice for upcoming industrial applications because of their potential for long-term cost reductions and environmental advantages.

## 6. Applications

The distinctive properties and advantages of biopolymer-based composites derived from agricultural waste biomass render them potentially useful in diverse industries. Below are several instances of their practical implementations [133]. Some of the applications are materials for building; boards, panels, and insulation may all be produced from composites made from biopolymers, which can then be utilized in construction. Because of their favorable mechanical qualities, fire resistance, and low thermal conductivity, these materials are appropriate for a wide variety of applications within the building industry. For components for automobiles, dashboards, door panels, and seats are examples of the types of car components that may be made with biopolymer-based composites. These materials offer great resistance to heat and chemicals, as well as strong mechanical qualities. In addition, these biopolymer-based composite's weight is quite low. For manufacture of biofuels, composites made from biopolymers have the potential to be utilized as feedstock in the manufacture of biofuels, such as bioethanol and biodiesel. These materials contain a high concentration of carbohydrates and are therefore important in a wide variety of conversion techniques that may be used to produce biofuels. Improvements in soil fertility, water retention, and nutrient availability can be attained by incorporating biopolymer-based composites into the soil. The soil and plants will benefit over the long term from the addition of these ingredients, which may be worked into the soil or used as a top dressing [134,135]. Biopolymer-based composites are utilized to create bone transplants, tissue scaffolds, and drug delivery systems, all of which have their place in modern medicine. The biocompatibility and biodegradability of these materials are favorable, thereby mitigating the potential for rejection and adverse reactions. The utilization of biopolymer-based composites derived from agricultural waste biomass exhibits a diverse array of potential applications across multiple industries [136–140]. These materials offer sustainable and environmentally conscious alternatives to conventional materials. Ultimately, it can be inferred that composites hold significant promise for future industrial use. Packaging, building, automobile parts, biofuels, soil improvement, and medical devices are a few examples of real-world items that have been successfully marketed using biopolymer-based composites [141]. A variety of feedstocks and packaging for biofuel products have been made using biopolymer-based composites in the biofuel industry. These materials have been demonstrated to be more environmentally friendly and sustainable than conventional materials, as well as being compostable and biodegradable. Biopolymer-based composites have been utilized to make biodegradable mulch films and soil additives for soil improvement. These substances are compostable and biodegradable and have been demonstrated to be useful in enhancing soil fertility and health. Biopolymer-based composites have been used in medical devices

to make a variety of implants and prostheses. These substances are perfect for use in medical applications because it has been demonstrated that they are biocompatible and biodegradable. The potential of these materials to replace conventional synthetic materials in a range of applications is generally shown by the real-world examples of goods that have been successfully marketed utilizing biopolymer-based composites [142–147].

## 7. Conclusions

In conclusion, biopolymer-based composites made from biomass obtained from agricultural waste present a sustainable and earth-friendly substitute for traditional materials in a variety of industries. These composites, which are created by mixing agricultural waste biomass and biopolymers, can be utilized for a variety of purposes, including packaging, construction materials, automobile parts, biofuels, soil improvement, and medical devices. These composites have the advantages of being affordable, renewable, and biodegradable, which can help to reduce the negative environmental effects of agricultural waste. However, creating these composites presents difficulties, including the unpredictability of agricultural waste biomass and the requirement for ideal processing conditions. Despite these difficulties, biopolymer-based composites have great promise as a material for future industrial use due to their prospective applications and advantages. The creation and application of biopolymer-based composites made from agricultural waste biomass will probably be crucial in helping society to achieve its goals of sustainability and environmental responsibility.

**Author Contributions:** All the authors contributed equally to the conceptualization, methodology, writing, reviewing, and editing of this manuscript. All authors have read and agreed to the published version of the manuscript.

**Funding:** Akarsh Verma would like to thank the University of Petroleum and Energy Studies, Dehradun, India (SEED Grant program) for the academic and financial support.

**Institutional Review Board Statement:** The authors hereby state that the present work is in compliance with the ethical standards.

**Data Availability Statement:** Not applicable.

**Conflicts of Interest:** The authors declare no conflict of interest.

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
