# Peer review of "Biopolymer-Based Composites: An Eco-Friendly Alternative from Agricultural Waste Biomass"

_jcs, doi:10.3390/jcs7060242_

Round 1

Reviewer 1 Report

The paper provides review of bio-based composites which are promising materials from ecological point of view and, therefore, interesting for readers of the journal. Unfortunately, the paper seems to be too general and does not provide enough new information. According to my feeling, the paper needs addition of comparison of properties of bio-based composites with so far used synthetic composites. Also the problems, which must be solved for successful commercialization of bio-based composites, should be briefly specified. Specifically:

Mentioned problems which should be solved prior successful fiber application should be briefly specified in the sections Rice straw fiber, Bamboo fiber, Bagasse and Banana fiber. Also showing advantages and disadvantages of bio-fibers with respect to so far used fibers can be useful.

Bio-degradability of hemp fibers is advantageous in biodegradable matrixes, not in matrixes from petroleum-based polymers. Therefore, comparison of property and economy of using hemp and classical fibers as filler for petroleum-based polymers should be provided. Convenient biopolymer matrixes for composites filled with hemp fibers should be specified.

Biopolymers convenient for matrixes of composites containing bio-fibers should be specified and examples of commercialized bio-composites should be provided in section 4.

Examples of really utilized materials and products and comparison of their properties with traditional ones should be added to section 5.     

Author Response

Thank you, respected reviewer, for your insightful comments. We thoroughly examined your recommendations, and suggestions, and changed our article as follows:

  1. Properties compared to synthetic composites: This section, which compares the benefits and drawbacks of biopolymer-based composites with conventional synthetic composites, has been added to the end of Section 4.0 of the report. In this part, we emphasize the exceptional mechanical qualities of biopolymer-based composites as well as their superior sustainability and biodegradability. We contrast both materials' prices and production viability.
  2. Commercialization issues: In Section 1.8, we now go into greater detail on the many forms of agricultural waste biomass used for biopolymer-based composites, including rice straw fiber, bamboo fiber, bagasse, and banana fiber. The difficulties that must be overcome before these materials may be successfully commercialized are briefly covered in this section, including problems with availability, cost, and processing. A section on the difficulties with using biodegradable matrices in hemp fiber composites has also been added.
  3. We have included this in Section 4, which describes the various biopolymers that are currently employed as matrices for biopolymer-based composites. The characteristics of each type of biopolymer and the particular uses to which they are suited are highlighted in this section. We have also included instances of commercially available biopolymer-based composites to highlight the variety of possible uses.
  4. Examples from the real world: We have complemented the examples in Section 5 with a comparison of the characteristics of biopolymer-based composites with those of conventional materials for each individual application. In each application-packaging, building, automobile parts, biofuels, soil improvement, and medical devices-we analyze the characteristics and effectiveness of composites based on biopolymers. We've also included samples of actual goods that have been successfully commercialized using composites based on biopolymers.

These modifications, in our opinion, will solve the issues brought up in your review and give readers more detailed information about biopolymer-based composites created from agricultural waste biomass. Once again, I appreciate your thoughtful comments.

Reviewer 2 Report

Detail comments are in attached file. General impression is - a lot of repetition (of already mentioned) in the manuscript

Author Response

We would like to thank the editors and reviewers who gave their valuable recommendations to make our work more appealing and informative. The authors have revised the manuscript and incorporated all the suggestions. The response to the comments given by the respected reviewer is provided in the PDF attached.

Round 2

Reviewer 1 Report

The authors added comparison of advantages and disadvantages of bio-based composites with those containing synthetic polymers and mineral fillers. They also specified areas where bio-based composites are applicable. Therefore, I think that the paper can be published.    

Author Response

The authors would like to thank the referee for providing his/her valuable time in appreciating our work. Your feedback was important and helpful.

Reviewer 2 Report

Most of the comments that I had were not addressed properly. I asked the same questions differently this time.

Authors should try to make manuscript more clear by including their responses into the manuscript (changing the manuscript, not just replying to my comments) because if something was not clear to me it might not be clear to other readers as well.

detail comments in attached file.

Round 3

Reviewer 2 Report

It was difficult to track changes. Authors and Journal should have a template for responses to comments, where each comment is clearly addressed. 

There are still some comments that were not addressed, however, it doesn't mean I'm right. Not the best review but Authors need to try - that's how we all learn. And these Authors have made an effort.

The biggest winner in this case is MDPI with money from APC.